# The Design and Validation of an Open-Palm Data Glove for Precision Finger and Wrist Tracking

**DOI:** 10.3390/s25020367

**Published:** 2025-01-09

**Authors:** Olivia Hosie, Mats Isaksson, John McCormick, Oren Tirosh, Chrys Hensman

**Affiliations:** 1School of Engineering, Swinburne University of Technology, Hawthorn, VIC 3122, Australia; lhosie@swin.edu.au; 2Centre for Transformative Media Technologies, Swinburne University of Technology, Hawthorn, VIC 3122, Australia; jmccormick@swin.edu.au; 3School of Health and Biomedical Sciences, Royal Melbourne Institute of Technology, Melbourne, VIC 3000, Australia; oren.tirosh@rmit.edu.au; 4School of Health Science, Swinburne University of Technology, Hawthorn, VIC 3122, Australia; 5College of Rehabilitation Sciences, Shanghai University of Medicine and Health Sciences, Shanghai 200237, China; 6Division of Robotics, Swinburne University of Technology, Hawthorn, VIC 3122, Australia; chensman@swin.edu.au; 7Department of Surgery, Monash University, Clayton, VIC 3800, Australia; 8Department of Medicine, University of Adelaide, Adelaide, SA 5005, Australia; 9LapSurgery Australia, Dandenong North, VIC 3175, Australia

**Keywords:** data glove, inertial measurement unit (IMU), open-palm design, resistive flex sensor (RFS), wearable technology

## Abstract

Wearable motion capture gloves enable the precise analysis of hand and finger movements for a variety of uses, including robotic surgery, rehabilitation, and most commonly, virtual augmentation. However, many motion capture gloves restrict natural hand movement with a closed-palm design, including fabric over the palm and fingers. In order to alleviate slippage, improve comfort, reduce sizing issues, and eliminate movement restrictions, this paper presents a new low-cost data glove with an innovative open-palm and finger-free design. The new design improves usability and overall functionality by addressing the limitations of traditional closed-palm designs. It is especially beneficial in capturing movements in fields such as physical therapy and robotic surgery. The new glove incorporates resistive flex sensors (RFSs) at each finger and an inertial measurement unit (IMU) at the wrist joint to measure wrist flexion, extension, ulnar and radial deviation, and rotation. Initially the sensors were tested individually for drift, synchronisation delays, and linearity. The results show a drift of 6.60°/h in the IMU and no drift in the RFSs. There was a 0.06 s delay in the data captured by the IMU compared to the RFSs. The glove’s performance was tested with a collaborate robot testing setup. In static conditions, it was found that the IMU had a worst case error across three trials of 7.01° and a mean absolute error (MAE) averaged over three trials of 4.85°, while RFSs had a worst case error of 3.77° and a MAE of 1.25° averaged over all five RFSs used. There was no clear correlation between measurement error and speed. Overall, the new glove design proved to accurately measure joint angles.

## 1. Introduction

Modern technology has made great strides in the area of precise upper-limb motion tracking, especially for the delicate motions of the fingers and wrists. Finger and wrist motion tracking is used in numerous fields, including surgical ergonomics and tool design, animation, augmented reality, animation, and rehabilitation. For finger and wrist tracking, a device should be compact and lightweight so that the hand can undergo full movement without restrictions.

Hand movements can be monitored through various techniques, including electromyography (EMG) wearables, optical tracking, and smart gloves. Advancements in wearable technologies, such as smart mechanoluminescent materials, have enabled the development of highly sensitive motion tracking systems that offer new opportunities for precise wrist tracking [1]. EMG sensors can be positioned around the wrist to monitor hand movements. This method permits the hand to move freely without constraints. Optical tracking utilises skin markers [2] or marker-less motion capture [3] for the wrist and fingers. Marker-less capture offers the advantages of unrestricted movement, as it does not obstruct the hands, enhancing user-friendliness. However, it frequently encounters challenges with occlusion, where objects or body parts block the line of sight between the cameras and the markers, leading to gaps or inaccuracies in the captured data. Other issues include intricate camera placements and long calibration processes [4]. Smart gloves can be classified into two categories: data gloves, which collect data and analyse movement only, and sensory gloves, which collect data but have been extended to also provide tactile or haptic feedback. Gloves facilitate direct and accurate measurement of the fingers and wrist; however, they frequently restrict natural hand movements or induce discomfort.

Various sensor types such as resistive flex sensors (RFSs), inertial motion units (IMUs) that combine accelerometers, gyroscopes and magnetometers, Hall-effect sensors, capacitive stretch sensors (CSSs), and optical sensors, are employed in smart gloves [5]. Several gloves incorporate IMUs to track motion, each encountering comparable limitations and outcomes. Moreira et al. developed an IMU glove that exhibits minimal inaccuracies and features a straightforward calibration process, ensuring that IMU gloves can provide accurate data [6]. Other IMU gloves have been developed to monitor hand functions for hand rehabilitation [7]. IMUs are typically used in parallel with other sensors such as CSSs and RFSs to provide a wider range of data [8]. CSSs are capable of measuring the elongation of a material by changing its capacitance when stretched. They are often used in wearable motion tracking technology due to their small size and ability to be attached directly onto the skin [9]. While they offer consistent performance, CSSs tend to require a larger input to produce noticeable output, making them slightly less sensitive than RFSs. RFSs have been utilised in numerous motion tracking gloves in the past. They are commonly employed alongside IMUs to collect more data on acceleration and wrist orientation as well as the flexion and extension of the joints [10,11,12,13]. Gloves combining RFSs and IMUs are commonly used in sign language translations [14].

The physical design of a smart glove can alter its characteristics and improve user-friendliness. Designs can be categorised into traditional gloves, a thimble attached to the fingertips, and exoskeletons with articulated structures over the hands. Traditional gloves can be closed-palm or fingerless, usually with a soft fabric design [5]. The more prevalent smart gloves are traditional closed-palm gloves that provide complete coverage for the hand and fingers [15]. A closed-palm glove presents numerous limitations, particularly in terms of movement restrictions. These constraints could be mitigated by employing an open-palm glove, which facilitates unimpeded hand movement and tactile feedback.

This paper validates a cost-effective, novel open-palm glove design utilising IMUs and RFSs. This open-palm glove design addresses the limitations of traditional motion capture gloves and enhances comfort and flexibility, making it well-suited for tasks requiring precise control and tactile feedback.

The reduced amount of fabric in the glove has several benefits for the fingers: there are less issues owing to hand shape, size variation, and the cost of manufacturing, and there is more comfort, as there is no fabric around the joints [16]. Unlike traditional gloves, which often make it difficult for users to insert their fingers into controllers or gadgets, this glove’s finger-free design allows for normal interaction with controllers. Many users can wear an open-palmed finger-free glove without the need for different sizes or variations for hands with various shapes and finger lengths. Many closed-palm gloves have sizing issues and can show data errors due to the glove’s fit [17]. Open-palm finger-free gloves provide users with greater control as the palm is free to feel the surface and detect the amount of surface friction and slipperiness. Friction-sensing ability is crucial to dexterity during tasks requiring precision gripping, such as when employing a two- or three-fingered grips [18]. These grips are commonly employed when using hand controllers, writing, using a computer, and performing many other everyday tasks. With friction-sensing, the user experiences less slippage, enabling the motion capture of tasks that demand a high level of precision, such as those in robotic surgery.

## 2. Related Works

Motion capture gloves have been available on the market for over 40 years, resulting in numerous validation studies on these devices. Commercially available motion capture gloves vary significantly in sensor types, applications, price range, and claimed accuracy [19]. The gloves listed below are the current state-of-the-art commercial gloves.

Rokoko Smart Gloves: Designed for VR applications, these gloves cost approximately AUD 1745 and utilise a combination of IMU and EMF sensors; they claim unmatched quality in hand and finger animation capture [20].

StretchSense Studio Glove: Designed for VR and AR applications, these gloves range from AUD 795 to AUD 5595 per pair. These gloves use stretch sensors and claim no occlusion or drift during motion capture [21].

Manus Motion Capture Gloves: Designed for film, animation, and live performance, these gloves combine IMU and RFSs for motion tracking and include haptic feedback sensors [22].

A common characteristic of these state-of-the-art gloves is their closed-palm design, which limits natural tactile feedback. While the fingertips are sometimes exposed, insufficient coverage remains for realistic haptic interaction. Other limitations of these gloves include limited validation under dynamic conditions, high costs, and the inconsistent reporting of critical specifications, such as sampling rates or motion range [5].

Academic research has explored alternatives to commercial gloves, focusing on different validation techniques and sensor designs. For instance, Lin et al. created a modular data glove using IMU and RFS sensors. Validation was performed in three steps: raw data validation, static angle verification, and dynamic angle verification. The raw data validation included comparing the IMU from the glove with a known reference IMU. Static angle verification involved connecting the sensors to a static angle verification instrument and comparing the measured angle to the real angle. The dynamic angle verification consisted of a small motor rotating through different predefined angles. Despite rigorous sensor testing, the glove itself was not validated in real-world conditions and covered the entire hand, limiting tactile feedback [23]. Moreover, Metcalf et al. validated a motion capture glove using a marker-less motion capture system. This involved comparing results to a ground truth estimation and a laboratory-based motion capture system [24]. Other academic motion capture gloves include an IMU-based glove proposed by Fang et al. that validated the dynamic conditions through different hand posture recognition [25]. Validation techniques for motion capture gloves vary widely, influenced by the intended application. While some studies prioritise sensor prevision, others emphasize overall glove performance, leaving gaps in comprehensive validation methods.

## 3. Materials

### 3.1. Glove Design

Our glove is fabricated from an elastic Lycra to allow for stretching during natural hand movements. Each finger of the glove has a 2.2-inch RFS sewn in, spanning from the proximal interphalangeal joint to the metacarpal phalangeal (MCP) joint, and extending towards the central metacarpal region of the hand. The sensors are sold by SparkFun electronics, Australia. As shown in Figure 1a, the glove has no fabric or wires covering the palmar surface of the hand or the finger tops. Each glove includes one IMU sensor, sewn onto the centre of the back of the hand with the y-axis aligned with the fingers, as seen in Figure 1b. There are no sensors devoted to the abduction and adduction of the fingers, as sensors between the fingers cause discomfort. To limit weight on the hand, the control box is attached to the elbow using Velcro. Whilst flat on the hand, the newly designed glove may still interfere with movements; however, placing sensors on the wrist only does not provide data on the finer finger and hand postures [26]. The overall cost for one glove is AUD 135.25; the breakdown of costs shown in Table 1.

#### 3.1.1. Inertial Measurement Units

Each glove incorporates one IMU to monitor the wrist flexion and extension. IMUs measure force through accelerometers, velocity through a combination of accelerometers and gyroscopes, and angular orientation through gyroscopes, and occasionally, magnetometers. Each sensor in the IMU has three degrees of freedom (DOFs) for the x, y, and z-axes. In most IMUs, there are six DOFs: three from the accelerometer measuring linear acceleration and three from the gyroscope measuring angular velocity. The addition of the magnetometer increases the total to nine DOFs and includes angular orientation data [27]. IMUs provide low-latency, high-frequency readings; however, because gyroscopes measure angular velocity and accelerometers measure linear acceleration, calculating orientation and angles requires integrating these measurements over time. This integration leads to drift due to the accumulation of tiny errors, making periodic calibration necessary. With the addition of the magnetometer, IMUs can measure absolute angles, offering a direct reading of orientation and hence not accumulating drift [28]. An IMU data glove must be calibrated to correct for stationary bias offset, which occurs when the accelerometer and gyroscope detect small, non-zero values for linear acceleration and angular velocity while the glove is at rest. This offset, which results from sensor bias, can build up over time and affect the accuracy of angle measurements [29]. In order to accommodate a wider range of applications, such as fine motion tracking, IMUs are designed to be compact in size [30]. When used for the purpose of joint tracking, IMUs must be placed tightly on the skin to prevent shifting during physical activity [31].

#### 3.1.2. Resistive Flex Sensors

The used RFSs measure flexion by altering their resistance, with values ranging between 1 kΩ to 20 kΩ depending on changes in the angle of deflection. RFSs are often made of special carbon ink and are flat, allowing them to conform to body joints [32]. RFSs can be used independently in gloves; however, the resistance must first be mapped to angles, as there is a slight nonlinearity over a large range of angles [33,34]. The resolution of an RFS is roughly 1° to 2° but varies dependent on the specific sensor and the voltage divider circuit used. The sensors provide discrete values of resistance for small changes in flexion and extension movements.

Using hybrid data from both IMUs and RFSs, the new glove determines joint angles of the MCP joints and the wrist allowing for accurate data collection.

### 3.2. Data Aquisition

A block diagram of the system electronics is shown in Figure 2. Each sensor is powered by a Raspberry Pi Pico wH microcontroller. The RFSs are connected in a voltage divider circuit, using a 380 Ω resistor for each sensor. As the microcontroller is small and only includes four ADC channels, an MCP30008 converter is used for the five RFSs. The MCP30008 is an analogue to digital converter that can connect directly to the microcontroller. The IMU, on the other hand, contains 16-bit analogue-to-digital conversion for each channel, allowing it to output digital data directly to the 3.3 V pin on the microcontroller without the need for an external converter. Each sensor was sampled at a sampling rate of 20 Hz. A program written in Thonny, an integrated development environment for Python development, was used to collect data. This data was later processed in MATLAB R2024a, version 24.1 (MathWorks, Natick, MA, USA) to convert raw flex sensor resistance and IMU acceleration and gyroscope values into angles using linear mapping and calibration.

### 3.3. Glove Evaluation Setup

The evaluation setup is shown in Figure 3. Autodesk’s Fusion 360 (Autodesk, San Rafael, CA, USA) was used to design a platform for the hand to rest on; the resulting design was 3D-printed. The hand testing robot interface was designed from an online model of the UR5e robot [35]. This robot was selected for its six DOF and precise motion control. The robotic system was controlled using the PolyScope software, version SW 10.7.0 (Universal Robots A/S, Odense, Denmark)., which allows for the seamless control of the robot. The robotic arm was mounted on a stable workbench and calibrated based on the manufacturer’s standard procedures to ensure accurate and repeatable movements. The 3D-printed design was made so that the rotation of the wrist only involves the movement on the fourth axis of the UR5e robot. This rotation axis is shown by the red line in Figure 4.

## 4. Methods

### 4.1. Sensor Testing

Prior to testing the glove, each sensor type was individually evaluated for drift, and the synchronisation between the two sensor types was assessed. To test the linearity of the resistive flex sensors (RFSs), the resistance was mapped to the angle of deflection for each sensor. Then, the results were compared to see if there were any variations between the sensors.

#### 4.1.1. Drift and Delay

The drift for each type of sensor was calculated by activating the sensor and allowing it to execute for one hour. This was repeated three times. In order to minimise errors, the sensors were not handled during this time. A synchronisation test was performed to determine any differences in delays between IMU and RFS values. In this test, both the RFSs and the IMU received a stimulus (rapid onset movement) at the same time. Both sensors can operate simultaneously as they are connected to the same microcontroller. The data and timestamps were collected and analysed using MATLAB. The real-time delay for each sensor was less than one sample (0.05 s). To ensure that long-term use is a possibility for this glove, a three-hour run test was conducted.

#### 4.1.2. Linear Mapping of the RFSs

The RFS were setup on top of a goniometer. As seen in Figure 5, the sensor was fastened to the goniometer with tape at each end but no tape in the middle to allow for bending. Initial resistance measurements were collected with the sensor in an unbent position. The goniometer was rotated until the resistance changed, then held for 5 s to record both the angle and resistance. This was repeated for the entire range of −90° to 90°. This process was performed for all five RFSs used in the glove.

### 4.2. Glove Testing

The individual was seated in a chair, with their elbow resting on the edge of a table and their forearm placed flat on the table. The chair was adjusted to a height that positioned the elbow at the most optimal and comfortable angle for the individual, as seen in Figure 3b. After the test subject donned the glove, it was ensured that the sensors were properly aligned with each knuckle. This experiment focused on the right hand. Prior to each test, the glove underwent calibration by being held in two easily attainable positions for 5 s each: flat and in a fist. The glove underwent calibration as an integrated system. The IMU was calibrated in situ on the hand to account for its placement and orientation within the glove, while the RFS sensors were simultaneously aligned using the pre-mapped resistance-to-angle correlation from above. This approach ensured that the entire system operated cohesively during motion tracking tests. In the flat position, the RFSs were at 0° flexion and the IMU at 0° and at the fist position, while the RFSs on the fingers were at 90° flexion; however, the thumb was placed at 45° of flexion. The calibration angles are shown in Figure 6.

#### 4.2.1. Static Measurements

Initially, a static analysis was conducted. The hand was positioned on the 3D-printed platform connected to the robot. A Velcro strap was used to fasten the hand, guaranteeing continuous contact with the robot throughout the recording. A visual check was conducted to ensure the hand remained flat on the 3D-printed platform, preventing any cupping that could alter the wrist angle.

When measuring the wrist angles, the hand was positioned with the fingers extended straight (0° flexion), and the thumb was positioned at a 30° angle to ensure comfort [36]. Fingers were measured in pairs with the remaining fingers lifted from the platform; the index and middle fingers were measured together, as well as the ring and little fingers. The angle was maintained for a duration of 10 s, with the central 6 s being documented using both the gloves and the robot. The angle was incrementally adjusted by 15° in both flexion and extension, with a maximum angle of 60° for both movements. The duration of each full test was 90 s, and the subject was provided with a 2-min break before repeating the test. To ensure user comfort, the thumb was only tested from −45° to 30°.

#### 4.2.2. Dynamic Measurements

After conducting the static testing, the glove’s dynamic capabilities were assessed. The robot spanned the angles from the maximum to minimal flexion of the wrist and each of the fingers, adopting the same setup as the static measurements. The robot employed different velocities to ensure that the glove could gather data over a spectrum of real-world motions. In dynamic testing, the angles of the fingers were only tested from −60° to 45° for user comfort and to mimic natural movements, where an extension above 45° is rare and painful. Three instrument velocities were evaluated: Slow (20°/s), Medium (25°/s), and Fast (30°/s).

#### 4.2.3. Data Analysis

The analysis began with data calibration to ensure accuracy and consistency across all measurements. Calibration involved adjusting the raw sensor outputs based on pre-determined calibration factors as explained above. For each trial, the worst-case error was calculated to quantify the maximum deviation observed, while the standard deviation (SD) was computed to measure the variability or spread of the data points around the mean for each measurement point. This process was conducted separately for IMU and RFS. The data from the trials were averaged to obtain a single representative data set for both IMU and RFSs. The next step involved evaluating the relationship between the sensor data and the real values (ground truth measurements). For this purpose, the coefficient of determination ($R^{2}$) was calculated for each sensor type. The $R^{2}$ value provides an indication of how well the sensor data fit the real values, with values closer to 1 indicating a better fit. The mean absolute error (MAE) was computed for each sensor. The MAE quantified the average magnitude of the errors between the sensor measurements and the real values, thus providing a measure of the overall accuracy of the sensors. An analysis of variance (ANOVA) was conducted to compare accuracy over the three angular velocities.

## 5. Results and Discussion

The primary aim of this study was to assess the measurement accuracy of angle measurements using a newly designed glove incorporating RFSs and IMUs, evaluated under static conditions at varying angular velocities.

Individual sensor drift tests were conducted for both the IMU and RFSs. The IMU had a drift of 6.60°/h, which is typical for low-cost sensors, although slightly lower than average drift for an MPU6050 [37]. As the drift was tested over a one-hour period, a drift could emerge over a longer period of time, though the glove is not designed for extended use without calibration and requires calibration every 15 min. A 15-min calibration interval was selected because the glove drifts approximately 1.65° within this time, which keeps the error within acceptable limits for accurate measurements while balancing practicality for users. The three-hour test showed a drift of 19.50° mirroring results from the one-hour test showing a drift of 6.5°/h. Following the drift testing, a synchronisation test was undertaken. This showed that on average, between all five RFSs and the IMU, there is only a 0.06 s delay, corresponding to one sample. While the study includes a range of angular velocities, even the fastest is not significantly affected by this minor delay, which is therefore disregarded in subsequent calculations.

The RFSs were calibrated by mapping resistance (Ω) to angle of deflection (°), as shown in Figure 7. Each RFS displayed a similar mapping of resistance to angle, but with different initial values, resulting in lines on the graph that appear almost parallel. The different initial values are as expected, as RFSs have different baseline resistances due to manufacturing. A line of best fit was found as well as the $R^{2}$ value, and is shown in Table 2. The RFSs performed similar to other RFSs described in the literature [38]. The sensor behaves linearly around 0° flexion, but as the flexion increases, the response becomes nonlinear, with the resistance changing less predictably at larger angles. Sensor variability is expected, particularly in low-cost sensors, which is why mapping each sensor separately is necessary. While the linear line of best fit is presented to assess how closely the data align with a linear trend, for glove calibration, the data were linearly extrapolated to achieve greater accuracy than what is provided by the line of best fit.

The evaluation of static angle measurements began by comparing the performance of both RFS and IMU sensors across the three trials. The worst case error for the RFSs was 3.76°. The SD found of the RFS measurements was 2.14°, indicating the variability of the data for each trial. For the IMU, the worst case error was 7.01°, with a SD of 2.16°. These similar errors and SD indicate that both sensors provide consistent measurements, though IMUs exhibited more variability. The variability in the IMU is due to the drift mentioned above. As all three trials were conducted within one calibration, the results drifted by roughly 1° over the 8-min period. Worst case error variability can be attributed to its continuous data acquisition over the angles from −60° to 60° in comparison to the discrete nature of the RFSs. The subsequent analysis was based on average values across all trials.

For the static data set, Figure 8 illustrates the IMU sensor results for the actual angles compared to the measured angles. The MAE and $R^{2}$ for IMU and RFSs are shown in Table 3. The IMUs demonstrated a high coefficient of determination for these data. This confirms a high level of linearity across the common range of motion (−60° to 60°). The MAE for the IMU sensors was low, showing a high accuracy in wrist angle measurements. The IMU sensors had a higher MAE than the RFS sensors; however, the error for the IMU has also been seen in other papers and matches other IMU-based data gloves error. For instance, when compared to the glove designed and validated by Connolly et al., their mean error for IMU sensors was 5.95° [39]. Hazman et al. similarly had a IMU glove with a static mean error of 5.41°, as did Mohan et al. with an IMU glove with a MAE of 6.3° for wrist flexion and extension [7,40].

Figure 9 and Figure 10 depict the static angle data for the fingers and thumb, respectively. For each data set, both the $R^{2}$ value and MAE were calculated to evaluate the sensor performance as shown in Table 3. The $R^{2}$ values for the five digits was averaged to 0.99, showing a high level of linearity across the range of −60° to 60° for the fingers and −45° to 30° for the thumb. The MAE for static finger conditions for the four fingers were generally lower than other RFS-based gloves; however, the thumb data were consistent with the other literature. For instance, in Gentner and Classen’s low-cost sensor glove, the overall error of the RFSs was reported as 4.96°, which is higher than the error found in this glove [41]. Other RFS-based gloves have errors of 3.4° [42].

Overall, the static measurements were highly accurate, with an average MAE of 1.22° across all MCP joints and the wrist. These results demonstrate the glove’s potential for precise angle measurement during static tasks.

To assess the glove’s performance in dynamic situations, the sensors were evaluated at three different angular velocities to simulate various motion scenarios. For each trial, the robot began at −60° and accelerated to the target velocities from −60° to −55°. Following this, the robot travelled at a constant speed from −55° to 55° before coming to a stop at 60°. Figure 11 exemplifies this motion over time for the IMU. For this experiment, the data were only collected from −55° to 55° as a constant speed is required to quantify the error between known and measured angles.

At each angular velocity, the MAE was computed as well as the $R^{2}$ values when plotted measured angle against time for both RFS and IMU sensors. These results are shown in Table 4.

To compare the differences between the performance at each angular velocity, a one-way ANOVA was conducted. The null hypothesis for this analysis is that there are no differences in the performance over the three angular velocities. The ANOVA test revealed that there were no statistical differences between the three velocities, F(2,9) = 0.08, *p* = 0.9219. In this case, we fail to reject the null hypothesis, suggesting that there is no statistically significant difference in performance over different angular velocities. This suggests that the angular velocities provide similar results indicating that the sensors will provide accurate angle measurements over any natural movement across many velocities.

The IMU performed the best results under dynamic conditions. The results maintained a high linearity $R^{2}$ and showed a low MAE for each velocity. There was minimal variation across the three different velocities. The RFSs also exhibited minimal variation across velocity, performing similarly at each. In contrary to static testing, the RFSs on average had a higher MAE than the IMUs. The MAE averaged for the glove (all sensors) was low: 4.98° (Slow), 4.85° (Medium), and 4.34° (Fast). These findings are similar to other RFS and IMU gloves in the literature, with IMU having a slightly better accuracy and reliability in dynamic movements when compared to RFSs [43].

Overall, the dynamic data, whilst not as accurate as the static data, showed results similar to the other literature over different velocities. For example, Oregno et al. showed a mean error of 3.7° in dynamic RFS measurements of joint angles [33]. The other literature shows similar data over different velocities [29].

It is important to note that the inconsistencies in RFS accuracy are likely due to sensor placement rather than angular velocity, underscoring the importance of proper glove alignment.

This study introduces and validates an innovative motion capture glove, although certain limitations must be recognised. This study largely concentrated on confirming the motion tracking accuracy of the glove, hence excluding evaluations of user comfort, convenience of use, or long-term wearability. These aspects will be examined in subsequent research. Furthermore, the study employed a restricted sample size, perhaps overlooking variations in hand dimensions, grip strengths, or dexterity levels. Ultimately, the glove underwent testing in a controlled environment, which may not entirely reflect its efficacy in real-world applications. Subsequent research should evaluate the glove in practical settings. To enhance the functionality of this glove, subsequent research should incorporate wireless connectivity. This will encompass the integration of the Internet of Things paradigm with the transmission of data in real-time. This advancement will enhance the glove’s utility, enabling the transmission of data to healthcare providers for tele-rehabilitation, thereby eliminating the necessity for in-person consultations. Further enhancements may involve incorporating force capture within the glove to collect a more extensive range of data.

## 6. Conclusions

Traditional closed-hand motion-capture gloves restrict tactile feedback by covering the entire hand. This study proposes and examines a new type of low-cost data glove that encompasses an open-palm design that allows for a high range of movement and friction sensing. The sensors were initially tested individually, followed by an evaluation of the glove’s performance over static and dynamic conditions for both RFS and IMU.

For static conditions, the glove demonstrated a high repeatability; the IMU had a worst case error across three trials of 7.01° and a mean absolute error (MAE) averaged over three trials of 4.85°, while RFSs had a worst case error of 3.77° and a MAE of 1.25° averaged over all five RFSs used. The average MAE of 1.85° across all MCP joints and the wrist indicates the high performance of the glove. The IMU sensors in static measurements had more errors than the RFSs; however, both MAEs were low and similar to other IMU and RFS data gloves. The dynamic conditions showed a marginally lower accuracy, but still provided sufficient detail to capture hand movements at three different angular velocites. There was little variation between the three velocities. Importantly, the ANOVA results confirmed no statistically significant differences between the angular velocities, further validating the glove’s consistency across varied motion rates. Whilst the average MAE across the angular velocities were higher than static conditions, the results still remained low: 4.98° (Slow), 4.85° (Medium), and 4.34° (Fast). Overall, the glove performed well and can accurately measure angles of the wrist and fingers over natural movement ranges across static and dynamic scenarios.

Future studies should ensure that the glove has optimal placement on the hand, as the results show that sensor displacement can affect performance. Additionally, exploring more natural hand movements rather than forced full-range motions would better reflect real-world use cases. The gloves have many possible applications that involve motion tracking while maintaining tactile feedback, such as in robotic surgery, sports, and other fields.

## Figures and Tables

**Figure 1 sensors-25-00367-f001:**
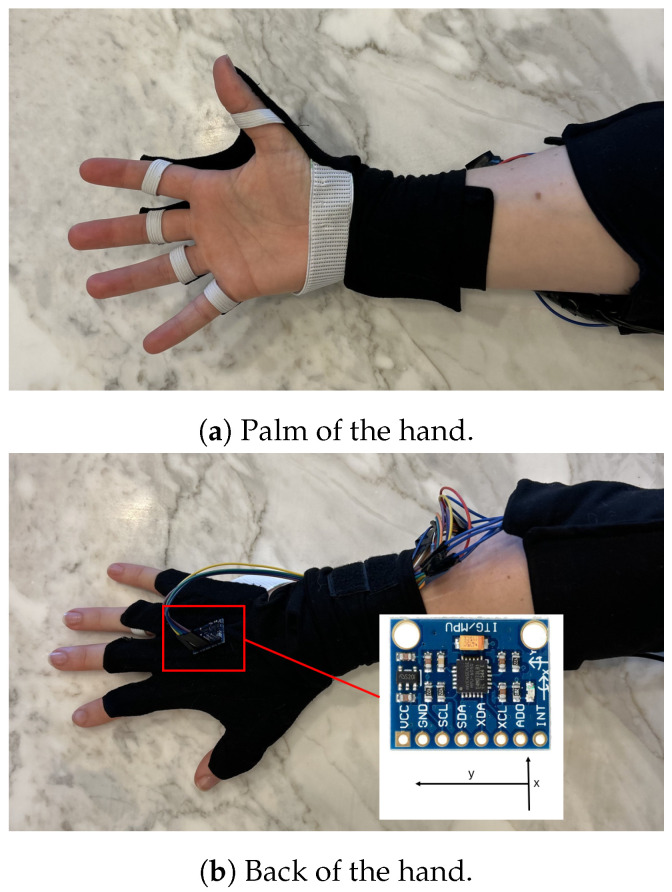
Images of the glove.

**Figure 2 sensors-25-00367-f002:**
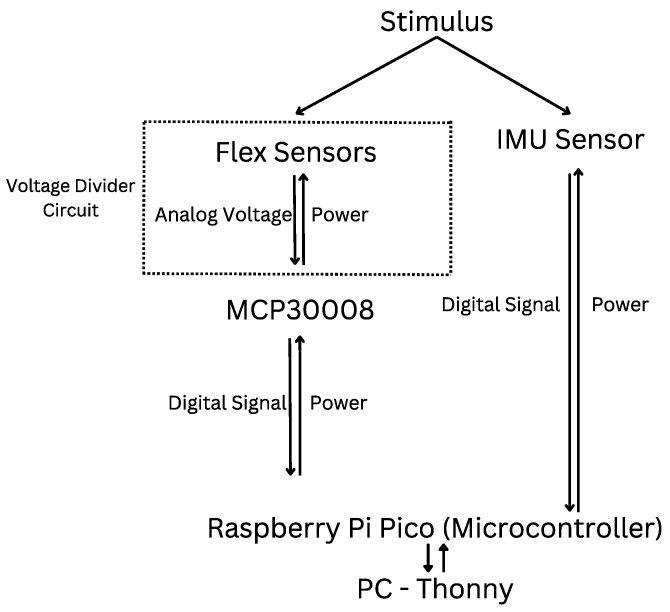
Block diagram of the system electronics.

**Figure 3 sensors-25-00367-f003:**
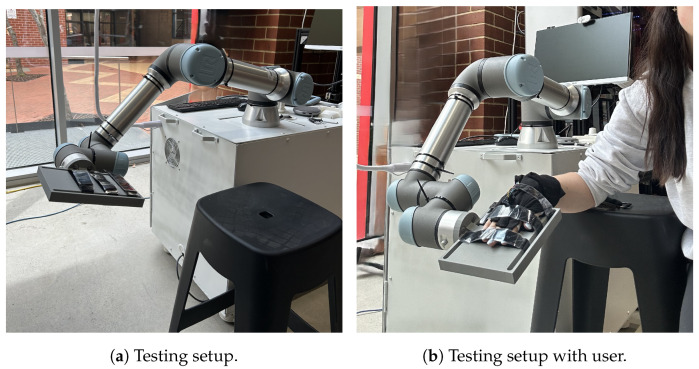
Robotic testing setup.

**Figure 4 sensors-25-00367-f004:**
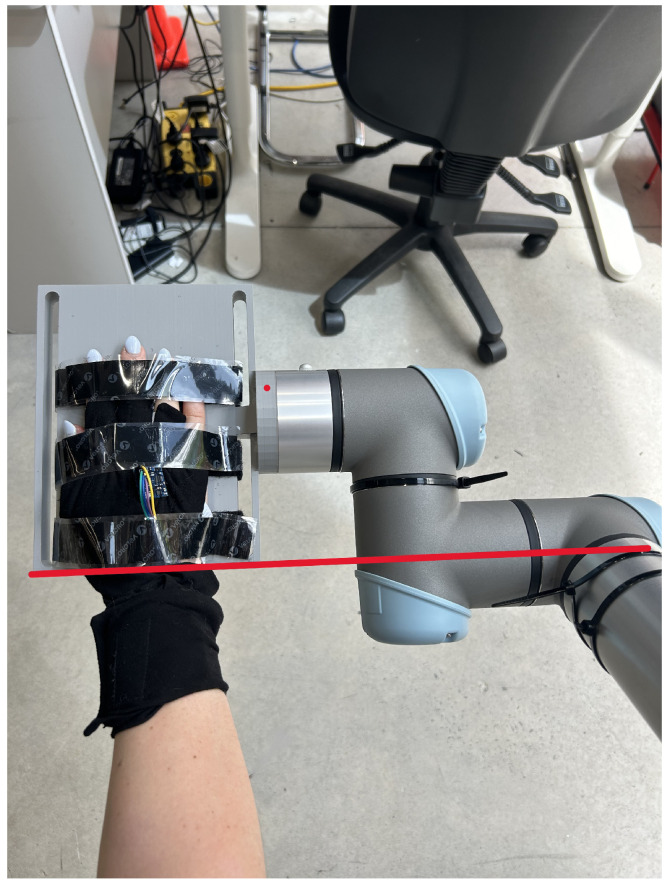
Hand in testing setup showing the glove placement. The red line shows the alignment of the fourth robot axis with the wrist joint.

**Figure 5 sensors-25-00367-f005:**
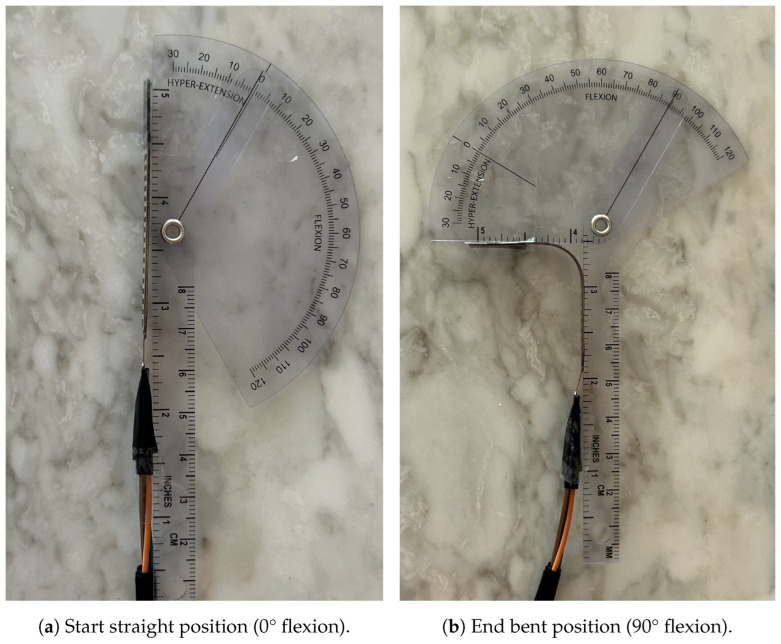
Start and end angles of linear mapping of RFSs.

**Figure 6 sensors-25-00367-f006:**
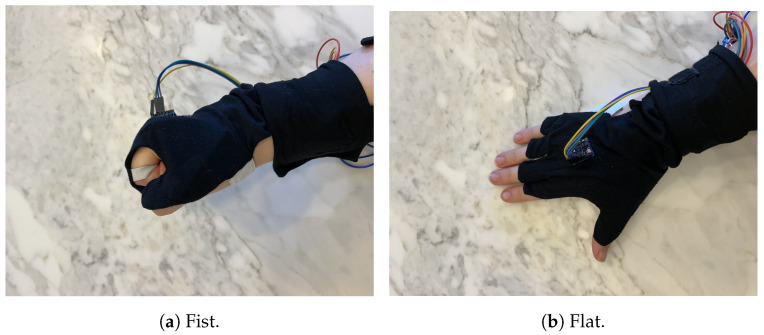
Calibration positions of hand.

**Figure 7 sensors-25-00367-f007:**
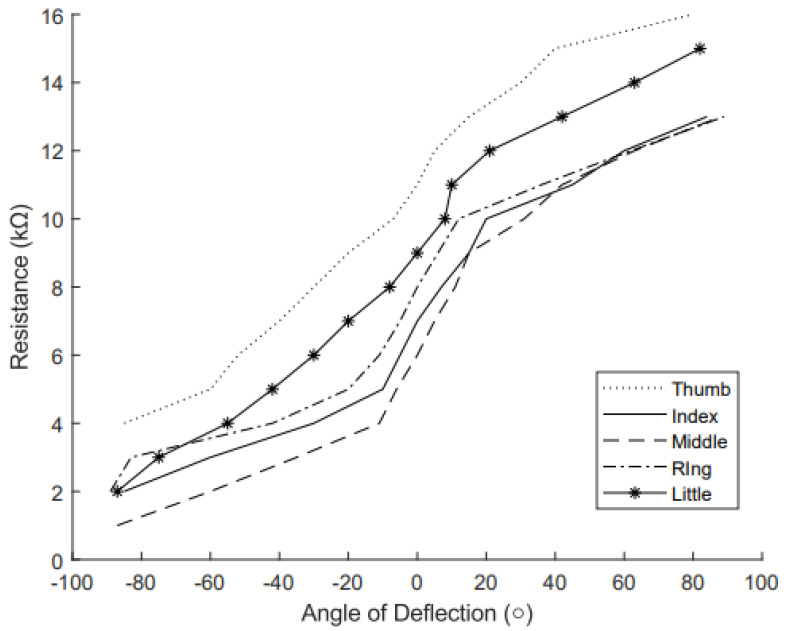
Mapping of the resistance (kΩ) to the angle of deflection (°). Each curve represents a different RFS used in the glove: dotted for thumb, solid for index, dashed for middle, dash-dotted for ring, and solid with stars for little finger. The graph shows the calibration of resistance to deflection angles, essential for accurate motion tracking.

**Figure 8 sensors-25-00367-f008:**
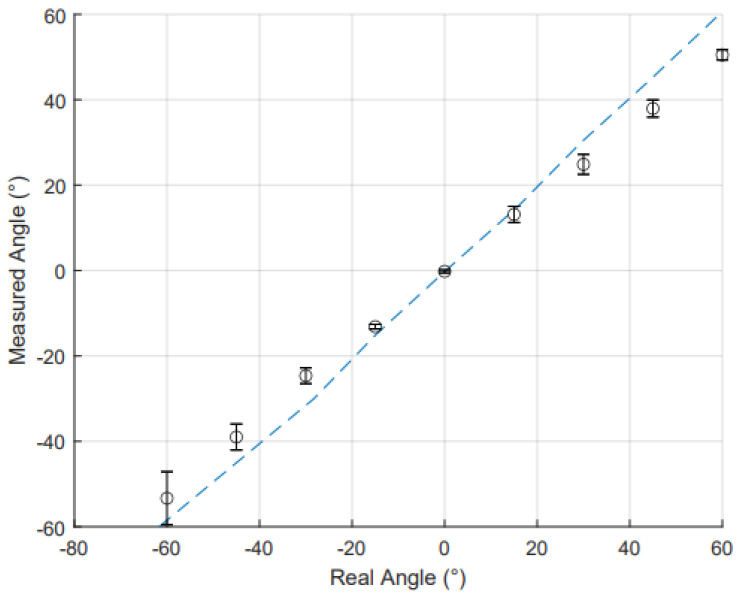
Comparison of measured wrist angle and real wrist angle as measured by IMU. The data points (o) represent the average value at each angle, with standard deviations across each trial shown by error bars. The dotted trendline shows the linearity of the relationship between measured and real angles. The close alignment of the data points with the trendline suggests minimal deviation, reinforcing the reliability of the IMU for wrist angle measurements.

**Figure 9 sensors-25-00367-f009:**
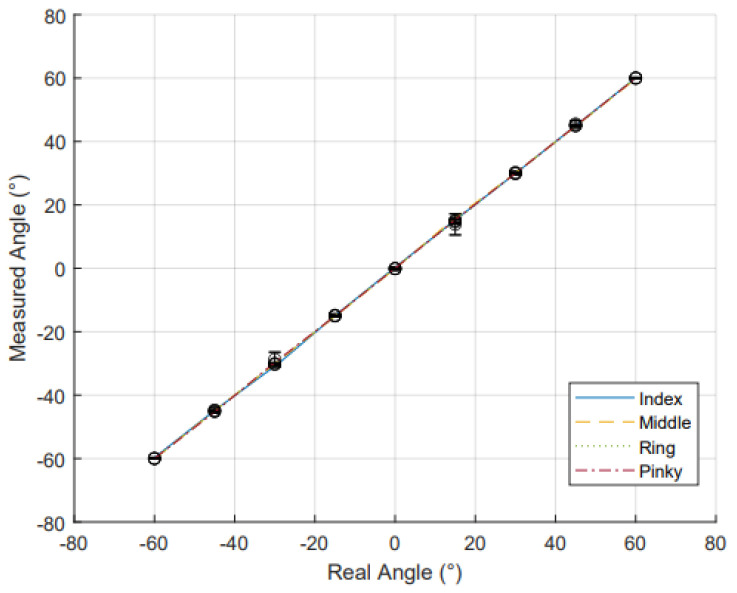
The comparison of the measured finger angle and real finger angle as measured by RFS. Data points (o) represent the average value at each angle, with standard deviations across each trial shown by error bars. The coloured dotted trendline shows the linearity of the relationship between measured and real angles for all the fingers: blue for index, yellow for middle, green for ring, and red for little finger.

**Figure 10 sensors-25-00367-f010:**
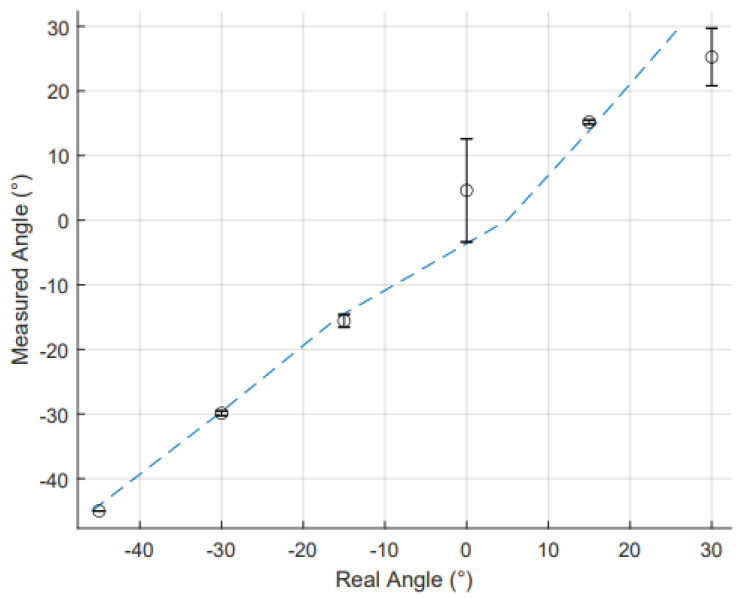
The comparison of measured thumb angle and real thumb angle as measured by RFS. Data points (o) represent the average value at each angle, with standard deviations across each trial shown by error bars. The dotted trendline shows the linearity of the relationship between measured and real angles.

**Figure 11 sensors-25-00367-f011:**
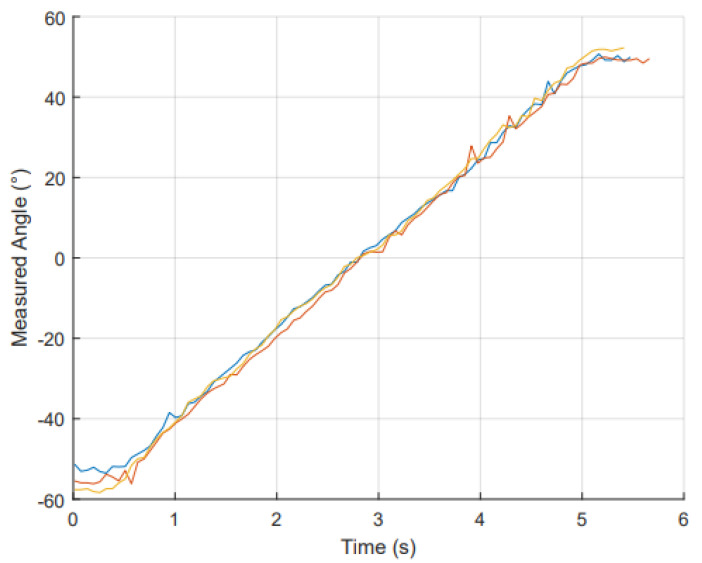
Progression of IMU angles (°) over a 6 s period, showing the change in angle as a function of time. The plot highlights the sensor’s responsiveness and consistency during continuous motion. Non-continuous motion at the start and end of the trial is included in this graph to demonstrate why it is excluded from further calculations.

**Table 1 sensors-25-00367-t001:** Table of costs for one glove (AUD).

Component	Unit Cost	Quantity Needed	Total Cost
Fabric	AUD 8 per metre	250 mm × 250 mm	$2
Elastic	AUD 6 per metre	100 mm strip	AUD 0.6
Velcro	AUD 12 per metre	100 mm strip	AUD 1.2
RFSs	AUD 19 each	5	AUD 95
IMU	AUD 3.75 each	1	AUD 3.75
Raspberry Pi Pico WH	AUD 11.5 each	1	AUD 11.5
MCP30008	AUD 8.7 each	1	AUD 8.7
380 Ω resistor	AUD 0.1 each	5	AUD 0.5
Breadboard with wire	AUD 6 each	2	AUD 12
		Total	AUD 135.25

**Table 2 sensors-25-00367-t002:** Linear regression equations modelling the relationship between resistance and angle. The ‘Linear Fit’ column shows equations of the form y=mx+b, where *y* is ther predicted angle, *x* is the resistance measured, *m* is the slope, and *b* is the intercept. The $R^{2}$ value indicates the goodness of fit for each equation reflecting the accuracy of the model for each finger.

RFS Placement	Linear Fit	$R^{2}$ of the Fit
thumb	12.87 x − 93.11	0.95
index	14.07 x − 109.30	0.94
middle	11.70 x − 105.90	0.98
ring	11.39 x − 123.4	0.97
little	11.82 x − 78.46	0.94

**Table 3 sensors-25-00367-t003:** MAE and $R^{2}$ values for each joint and sensor type under static conditions. The MAE values, reported with 95% confidence intervals (CI), quantify the average angular error between measured and real angles. The $R^{2}$ value indicates the goodness of fit for the linear relationship between measured and actual angles.

Sensor Type	Joint	MAE [95% CI] (°)	$R^{2}$
IMU	Wrist	4.85 [4.78, 4.91]	0.99
RFS	Index	0.28 [0.27, 0.29]	0.98
RFS	Middle	0.19 [0.19, 0.20]	0.97
RFS	Ring	0.13 [0.13, 0.14]	0.99
RFS	Little	0.14 [0.14, 0.15]	0.97
RFS	Thumb	1.71 [1.64, 1.77]	0.90

**Table 4 sensors-25-00367-t004:** Comparison of $R^{2}$ and MAE values for each joint and sensor type over three angular velocities: Slow, Medium, and Fast. MAE and $R^{2}$ values for each joint and sensor type under static conditions. The MAE values, reported with 95% confidence intervals (CI), quantify the average angular error between measured and real angles. The $R^{2}$ value indicates the goodness of fit for the linear relationship between measured and actual angles.

$R^{2}$ Values				
Sensor Type	Joint	Slow Speed	Medium Speed	Fast Speed
IMU	Wrist	0.999	0.999	0.999
RFS	Index	0.978	0.979	0.982
RFS	Middle	0.959	0.966	0.972
RFS	Ring	0.980	0.989	0.964
RFS	Little	0.978	0.967	0.981
RFS	Thumb	0.959	0.903	0.972
**Mean Absolute Error Averages °**				
**Sensor Type**	**Joint**	**Slow Speed**	**Medium Speed**	**Fast Speed**
IMU	Wrist	3.933	4.052	3.668
RFS	Index	5.273	4.134	4.787
RFS	Middle	5.006	4.450	4.143
RFS	Ring	4.632	3.762	4.862
RFS	Little	5.603	5.468	4.632
RFS	Thumb	5.409	5.955	4.067

## Data Availability

The data analyzed during the current study are available upon reasonable request from the corresponding author.

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
