# Peer review of "The Design and Validation of an Open-Palm Data Glove for Precision Finger and Wrist Tracking"

_sensors, 2025, doi:10.3390/s25020367_

Round 1
Reviewer 1 Report
Comments and Suggestions for Authors
The authors describe a glove designed to monitor wrist and finger movements. The glove uses an inertial measurement unit and resistive flex sensors. To improve comfort and effectiveness for applications like robotic surgery, the glove aims to solve the limitations of traditional gloves. A cobot is used for simulations, and additional customised tests are used to evaluate performance. In static tests, the glove is attached to a platform, joint angles are changed, and data is recorded. Three angular velocities of natural hand movements are replicated in dynamic experiments. MATLAB is used to analyse data using metrics such as coefficient of determination and mean absolute error.
Although the work presents a good level of novelty, as the proposed glove appears valid, the paper requires improvements due to some issues. The main problems are described as follows:
- The paper needs an improved discussion of the state of the art, both in academic and commercial contexts. I suggest adding a section that examines better the existing solutions and limitations.
- A more detailed explanation of the setup (robotic system, software, and hardware configuration) is needed. While the authors describe the testing procedures, the details regarding the simulation setup are sparse.
- Other tests could improve the simulations. Suggested additional simulations can include long-term performance tests and testing the glove on diverse participants. Others can be performed in scenarios that simulate real-world applications.
- The glove design is presented as more comfortable than traditional ones. However, the paper does not include user feedback or other qualitative assessments.
Author Response
Dear Reviewer,
Thank you for taking the time to review our paper, “Design and Validation of an Open-Palm Data Glove for Precision Finger and Wrist Tracking”, and providing your valuable feedback. Your comments and suggestions have been instrumental in improving the quality and clarity of the work. We have carefully considered each point raised and have provided detailed responses below.
Please see below our point-by-point detailed response to your comments.
We greatly appreciate your insights and hope that our responses and revisions address your concerns satisfactorily. Please do not hesitate to let us know if further clarification is required.
Kind regards,
Olivia Hosie
School of Engineering,
Swinburne University of Technology,
Hawthorn Campus,
VIC 3122, Australia
Email: lhosie@swin.edu.au
Tel: +61 421 815 716
REVIEWER 1
Comment 1: “The paper needs an improved discussion of the state of the art, both in academic and commercial contexts. I suggest adding a section that examines better the existing solutions and limitations.”
Response: Thank you for your feedback. In response to your suggestion, we have added a new section titled “Related Works” to provide a more thorough discussion of the state-of-the-art in both academic and commercial contexts. This section examines existing solutions, their applications, and associated limitations. We have also highlighted how our work builds upon or diverges from these existing approaches.
The included content is as follows (Page 3):
“Motion capture gloves have been available on the market for over 40 years, resulting in numerous validation studies on these devices. Commercially available motion capture gloves vary significantly in sensor types, applications, price range and claimed accuracy [19]. Gloves listed below are the current state-of-the-art commercial gloves.
Rokoko Smart Gloves: Designed for VR applications, these gloves cost approximately 1745 AUD and utilise a combination of IMU and EMF sensors, they claim unmatched quality in hand and finger animation capture [19].
StretchSense Studio Glove: Designed for VR and AR applications, these gloves range from 795 AUD to 5595 AUD per pair. These gloves use stretch sensors and claim no occlusion or drift during motion capture [20].
Manus Motion Capture Gloves: Designed for film, animation, and live performance, these gloves combine IMU and RFSs for motion tracking and include haptic feedback sensors [21].
A common characteristic of these state of the art gloves is their closed-palm design, which limits natural tactile feedback. While fingertips are sometimes exposed, insufficient coverage remains for realistic haptic interaction. Other limitations of these gloves include limited validation under dynamic conditions, high costs, and inconsistent reporting of critical specifications such as sampling rates or motion range [4]. Academic research has explored alternatives to commercial gloves, focusing on different validation techniques and sensor designs. For instance, Lin et al. created a modular data glove using IMU and RFS sensors. Validation was performed in three steps: raw data validation, static angle verification and dynamic angle verification. The raw data validation included comparing the IMU from the glove with a known reference IMU. Static angle verification involved connecting the sensors to a static angle verification instrument and comparing the measured angle to the real angle. The dynamic angle verification consisted of a small motor rotating through different predefined angles. Despite rigorous sensor testing, the glove itself was not validated in real-world conditions and covered the entire hand, limiting tactile feedback [22]. Moreover, Metcalf et al. validated a motion capture glove using a markerless motion capture system. This involved comparing results to a ground truth estimation and a laboratory-based motion capture system [23]. Other academic motion capture gloves include an IMU-based glove proposed by Fang et al. that validated the dynamic conditions through different hand posture recognition [24]. Validation techniques for motion capture gloves vary widely, influenced by the intended application. While some studies prioritise sensor prevision, others emphasize overall glove performance, leaving gaps in comprehensive validation methods.”
Comment 2: “A more detailed explanation of the setup (robotic system, software, and hardware configuration) is needed. While the authors describe the testing procedures, the details regarding the simulation setup are sparse.”
Response: Thank you for the comment. In response, we have updated the paragraph on the testing set up itself detailing the robotic system software and hardware configuration as well as more details regarding the simulation set up.
The included content is as follows (Page 6):
“The evaluation setup is shown in Fig. 3. Autodesk's Fusion 360 (Autodesk, San Rafael, CA) was used to design a platform for the hand to rest on, the resulting design was 3D printed. The hand testing robot interface was designed from an online model of the UR5e robot [34]. This robot was selected for its six DOF and precise motion control. The robotic system was controlled using the PolyScope software, which allows seamless control of the robot. The robotic arm was mounted on a stable workbench and calibrated based on the manufacturer's standard procedures to ensure accurate and repeatable movements. The 3D printed design was made so that rotation of the wrist only involves movement of the fourth axis of the UR5e robot. This rotation axis is shown by the red line in Fig. 4.”
Comment 3: “Other tests could improve the simulations. Suggested additional simulations can include long-term performance tests and testing the glove on diverse participants. Others can be performed in scenarios that simulate real-world applications.”
Response: Thank you for your suggestion. We appreciate your recommendation to include additional simulations, such as long-term performance tests, testing on diverse participants, and real-world application scenarios. These are important areas for further exploration and align well with our vision for extending this research. In response, we will incorporate a three-hour continuous use test in the revised manuscript to provide insights into the glove's mid-term performance. This addition will enhance the current study's scope by offering preliminary data on its durability and consistency over extended use. We also acknowledge the importance of longer-term testing, diverse participant trials, and real-world applications. These aspects are beyond the scope of the present study due to time and resource constraints. However, we are planning to address them comprehensively in subsequent research, as they are integral to validating the glove's usability in practical and varied environments.
The included content is as follows (Page 6 and 9):
“To ensure long-term use is a possibility for this glove, a three hour run test was conducted.”
“The three hour test showed a drift of 19.50° mirroring results from the one hour test showing a drift of 6.5°/h”
Comment 4: “The glove design is presented as more comfortable than traditional ones. However, the paper does not include user feedback or other qualitative assessments.”
Response: Thank you for bringing this to our attention. We agree that qualitative assessments and user feedback would provide valuable insights into the glove’s comfort compared to traditional designs. However, the primary focus of this paper is on validating the glove’s motion tracking capabilities. As such, incorporating user feedback falls outside the scope of the current study. That said, we fully recognize the significance of evaluating user comfort and plan to address this comprehensively in a subsequent study. This future work will focus on user experiences, including comfort assessments, while performing various real-life tasks with different hand controllers.
Reviewer 2 Report
Comments and Suggestions for Authors
Hosie and coworkers report an interesting work that an open-palm data glove for precision finger and wrist tracking. The glove is potential for a variety of uses, including robotic surgery, rehabilitation, and so on. The paper could be published after the following minor consideration,
1. The results presented could be improved, for example, each Figure/Table caption is too simple, for readers, it is difficult to obtain sufficient information, it could be described in details.
2. To demonstrate the repeatability and reliability of the data, the error bars are recommended, as for Figure 7.
3. Pls adding some discussions on the capture of motion speed and pressure/force by the designed glove. Pls discuss the environmental adaptability of gloves, and compare their performance with other gloves to highlight their advantages
4. As mentioned, “Hand movements can be monitored through various techniques, including electromyo-graphy (EMG) wearables, optical tracking, and smart gloves...” I also agree with this part of the discussion. As enhance the citation section, some new references from several pertinent literature sources are recommended, for example: Science Bulletin,2021, 66 (3), 206-209; Responsive Materials, 2024, 2 (3), e20240019; Advanced Science, 2023, 10 (3), 2204925.
Author Response
RESPONSE TO REVIEWERS
Dear Reviewer,
Thank you for taking the time to review our paper, “Design and Validation of an Open-Palm Data Glove for Precision Finger and Wrist Tracking”, and providing your valuable feedback. Your comments and suggestions have been instrumental in improving the quality and clarity of the work. We have carefully considered each point raised and have provided detailed responses below.
Please see below our point-by-point detailed response to your comments.
We greatly appreciate your insights and hope that our responses and revisions address your concerns satisfactorily. Please do not hesitate to let us know if further clarification is required.
Kind regards,
Olivia Hosie
School of Engineering,
Swinburne University of Technology,
Hawthorn Campus,
VIC 3122, Australia
Email: lhosie@swin.edu.au
Tel: +61 421 815 716
Comment 1: The results presented could be improved, for example, each Figure/Table caption is too simple, for readers, it is difficult to obtain sufficient information, it could be described in details.
Response: Thank you for your insightful feedback. We appreciate your suggestion to provide more detailed Figure and Table captions to enhance the clarity and informativeness of the results. In response, we have revised most of the captions throughout the manuscript to include additional context, such as the experimental setup, data significance, and key findings associated with each figure or table. These changes aim to provide readers with sufficient information to interpret the results more effectively without requiring extensive cross-referencing within the text.
The revised content is as follows:
“Figure 7: Mapping of the resistance (kΩ) to the angle of deflection (°). Each curve represents a
different RFS used in the glove: dotted for thumb, solid for index, dashed for middle, dash-dotted for
ring and solid with stars for little finger. The graph shows the calibration of resistance to deflection
angles, essential for accurate motion tracking.”
“Figure 8: Comparison of measured wrist angle and real wrist angle as measured by IMU. Data points (o) represent the average value at each angle, with standard deviations across each trial shown by error bars. The dotted trendline shows the linearity of the relationship between measured and real angles. The close alignment of the data points with the trendline suggests minimal deviation, reinforcing the reliability of the IMU for wrist angle measurements.”
“Table 3: MAE and R2 values for each joint and sensor type under static conditions. The MAE values, reported with 95% confidence intervals (CI), quantify the average angular error between measured and real angles. The R2 value indicates the goodness of fit for the linear relationship between measured and actual angles.”
“Figure 11: Progression of IMU angles (⁰) over a 6 s period, showing the change in angle as a function of time. The plot highlights the sensor's responsiveness and consistency during continuous motion. Non-continuous motion at the start and end of the trial is included in this graph to demonstrate why it is excluded from further calculations.”
“Table 4: Comparison of R2 and MAE values for each joint and sensor type over three angular velocities: Slow, Medium, and Fast. MAE and R2 values for each joint and sensor type under static conditions. The MAE values, reported with 95% confidence intervals (CI), quantify the average angular error between measured and real angles. The R2 value indicates the goodness of fit for the linear relationship between measured and actual angles”
Comment 2: To demonstrate the repeatability and reliability of the data, the error bars are recommended, as for Figure 7.
Response: Thank you for your feedback and for highlighting the importance of error bars. For this particular graph, error bars are not included because the data presented is based on a single trial for each sensor across the range of angles. As such, there was no repeated testing to calculate variability or standard error. However, I appreciate your suggestion and will consider incorporating error bars in future graphs where repeatability and reliability are assessed through multiple trials.
Comment 3: Pls adding some discussions on the capture of motion speed and pressure/force by the designed glove. Pls discuss the environmental adaptability of gloves, and compare their performance with other gloves to highlight their advantages
Response: Thank you for your valuable feedback. Regarding environmental adaptability, the open-palm design of the glove allows for greater flexibility and tactile feedback, making it suitable for diverse environments and applications where unrestricted movement is critical. We have now updated the discussion to better highlight how the glove's performance compares to traditional gloves, emphasizing its advantages such as improved range of motion and comfort. Additionally, while this study focuses on motion angle capture, I have clarified the glove’s current capabilities regarding motion speed and pressure/force capture, noting these as areas for potential future enhancement. I appreciate your suggestions and believe these changes will strengthen the paper.
The included content is as follows (Pages 10, 11 and 14):
“For instance, when compared to the glove designed and validated by Connolly et al. their mean error for IMU sensors were 5.95° [38]. Hazman et al. similarly had a IMU glove with a static mean error of 5.41° as did Mohan et al. with an IMU glove with a MAE of 6.3° for wrist flexion and extension [39, 6].”
“The MAE for static finger conditions for the 4 fingers were generally lower than other RFS based gloves, however, the thumb data was consistent with other literature. For instance, in Gentner and Classen’s low-cost sensor glove, the overall error of the RFSs was reported as 4.96° which is higher than the error found in this glove [40]. Other RFS-based gloves have errors of 3.4°.”
“Further enhancements may involve incorporating force capture within the glove to collect a more extensive range of data.”
Comment 4: As mentioned, “Hand movements can be monitored through various techniques, including electromyography (EMG) wearables, optical tracking, and smart gloves...” I also agree with this part of the discussion. As enhance the citation section, some new references from several pertinent literature sources are recommended, for example: Science Bulletin,2021, 66 (3), 206-209; Responsive Materials, 2024, 2 (3), e20240019; Advanced Science, 2023, 10 (3), 2204925.
Response: Thank you for your agreement and for suggesting additional references. I appreciate the recommendation to enhance the citation section. We have incorporated relevant findings to strengthen the introduction and discussion sections. These references will help provide a broader perspective on techniques for monitoring hand movements and the advancements in wearable and motion-tracking technologies
The included content is as follows (Page 1):
“Advancements in wearable technologies, such as smart mechanoluminescent materials, have enabled the development of highly sensitive motion tracking systems that offer new opportunities for precise wrist tracking [1].”
Reviewer 3 Report
Comments and Suggestions for Authors
I would like to thank the authors for their efforts in producing this work. The paper reads well and brings something to the field of research. However, I have a few comments that should be reconsidered in order to improve the quality of your paper.
Please add the related work section.
A comparison with other similar work should be added.
Please add an independent discussion section and present the limitations of your study.
Has the glove been tested for durability over extended periods?
Was any calibration performed for the IMU and RFS before testing?
How do the reported errors (e.g., MAE for the IMU and RFSs) compare to commercially available gloves
What improvements are planned for glove design? For example, you could include the Iot concept, which will be beneficial for tele-rehabilitation.
Comments on the Quality of English LanguageSome typos should be corrected for example in the abstract: ''It is be especially...." ??
Some sentences could be more concise. For example, "The glove’s performance was tested with a novel angle testing setup mounted on a collaborative robot" might be shortened
Author Response
Dear Reviewer,
Thank you for taking the time to review our paper, “Design and Validation of an Open-Palm Data Glove for Precision Finger and Wrist Tracking”, and providing your valuable feedback. Your comments and suggestions have been instrumental in improving the quality and clarity of the work. We have carefully considered each point raised and have provided detailed responses below.
Please see below our point-by-point detailed response to your comments.
We greatly appreciate your insights and hope that our responses and revisions address your concerns satisfactorily. Please do not hesitate to let us know if further clarification is required.
Kind regards,
Olivia Hosie
School of Engineering,
Swinburne University of Technology,
Hawthorn Campus,
VIC 3122, Australia
Email: lhosie@swin.edu.au
Tel: +61 421 815 716
REVIEWER 3
Comment 1: “Please add the related work section.”
Response: Thank you for your feedback. In response to your suggestion, we have added a new section titled “Related Works”. This section examines existing solutions, their applications, and associated limitations. We have also highlighted how our work builds upon or diverges from these existing approaches.
The included content is as follows (Page 3):
“Motion capture gloves have been available on the market for over 40 years, resulting in numerous validation studies on these devices. Commercially available motion capture gloves vary significantly in sensor types, applications, price range and claimed accuracy [18]. For example:
Rokoko Smart Gloves: Designed for VR applications, these gloves cost approximately 1745 AUD and utilise a combination of IMU and EMF sensors, they claim unmatched quality in hand and finger animation capture [19].
StretchSense Studio Glove: Designed for VR and AR applications, these gloves range from 795 AUD to 5595 AUD per pair. These gloves use stretch sensors and claim no occlusion or drift during motion capture [20].
Manus Motion Capture Gloves: Designed for film, animation, and live performance, these gloves combine IMU and RFSs for motion tracking and include haptic feedback sensors [21].
A common characteristic of these state of the art gloves is their closed-palm design, which limits natural tactile feedback. While fingertips are sometimes exposed, insufficient coverage remains for realistic haptic interaction. Other limitations of these gloves include limited validation under dynamic conditions, high costs, and inconsistent reporting of critical specifications such as sampling rates or motion range [4]. Academic research has explored alternatives to commercial gloves, focusing on different validation techniques and sensor designs. For instance, Lin et al. created a modular data glove using IMU and RFS sensors. Validation was performed in three steps: raw data validation, static angle verification and dynamic angle verification. The raw data validation included comparing the IMU from the glove with a known reference IMU. Static angle verification involved connecting the sensors to a static angle verification instrument and comparing the measured angle to the real angle. The dynamic angle verification consisted of a small motor rotating through different predefined angles. Despite rigorous sensor testing, the glove itself was not validated in real-world conditions and covered the entire hand, limiting tactile feedback [22]. Moreover, Metcalf et al. validated a motion capture glove using a markerless motion capture system. This involved comparing results to a ground truth estimation and a laboratory-based motion capture system [23]. Other academic motion capture gloves include an IMU-based glove proposed by Fang et al. that validated the dynamic conditions through different hand posture recognition [24]. Validation techniques for motion capture gloves vary widely, influenced by the intended application. While some studies prioritise sensor prevision, others emphasize overall glove performance, leaving gaps in comprehensive validation methods.”
Comment 2: “A comparison with other similar work should be added.”
Response: Thank you for bringing this to our attention. As above we have added a related works section to this study. In this section we have included a comprehensive comparison of our work with other similar studies. This comparison spans both commercially available motion capture gloves and academic research.
For instance, we examine the limitations of existing gloves, such as closed-palm designs and inconsistent reporting of specifications, and compare these to our glove’s open-palm design. Additionally, the section details various validation methods used in prior studies (e.g., static, dynamic, and real-world validations) and evaluates how our approach builds upon these methods to address identified gaps.
Comment 3: “Please add an independent discussion section and present the limitations of your study.”
Response: Thank you for this valuable suggestion. To address this, we have added a limitations section to the manuscript, which transparently outlines the constraints of our study and suggests areas for future research. This section discusses aspects such as the focus on motion tracking and not user satisfaction, restrictive sample sizes and controlled environments. Additionally, we have expanded the Results section to include a comparison of the MAE values obtained in this study with known values from similar works. This addition provides a clear context for the accuracy of our system and highlights its contributions relative to existing solutions.
The included content is as follows (Page 14):
“This study introduces and validates an innovative motion capture glove, although certain limitations must be recognised. This study largely concentrated on confirming the motion tracking accuracy of the glove, hence excluding evaluations of user comfort, convenience of use, or long-term wearability. These aspects will be examined in subsequent research. Furthermore, the study employed a restricted sample size, perhaps overlooking variations in hand dimensions, grip strengths, or dexterity levels. Ultimately, the glove underwent testing in a controlled environment, which may not entirely reflect its efficacy in real-world applications. Subsequent research should evaluate the glove in practical settings.”
Comment 4: “Has the glove been tested for durability over extended periods?”
Response: Thank you for your feedback. Yes, the glove has been tested over an hour as it listed in the “drift” section of the paper. However, from your comment we believe it should be tested over a longer period of time. In response, we will incorporate a three-hour continuous use test in the revised manuscript to provide insights into the glove's mid-term performance. This addition will enhance the current study's scope by offering preliminary data on its durability and consistency over extended use. We also acknowledge the importance of longer-term testing, diverse participant trials, and real-world applications. These aspects are beyond the scope of the present study due to time and resource constraints. However, we are planning to address them comprehensively in subsequent research, as they are integral to validating the glove's usability in practical and varied environments.
The included content is as follows (Page 6 and 9):
“To ensure long-term use is a possibility for this glove, a three hour run test was conducted.”
“The three hour test showed a drift of 19.50° mirroring results from the one hour test showing a drift of 6.5°/h”
Comment 5: “Was any calibration performed for the IMU and RFS before testing?
Response: We are grateful for your constructive input. In our study, calibration was performed at the glove level, with both the IMU and RFS sensors calibrated as part of the integrated glove system. While the sensors were tested individually prior to integration to ensure proper functionality, they were not pre-calibrated outside the glove. Instead, calibration was performed collectively once the sensors were incorporated into the glove. Specifically, the RFS sensors were mapped prior to glove assembly to correlate resistance values with angular displacements. This mapping allowed for precise alignment of resistances to angles during glove calibration. The glove-level calibration involved both the IMU and RFSs calibrated simultaneously while the glove was on the hand. This ensured alignment and accuracy during real-world use.
To clarify this in the manuscript, we will revise the text to better describe the calibration process, emphasizing that sensor calibration occurs within the context of the integrated glove rather than as standalone procedures.
The included content is as follows (Page 7):
“The glove underwent calibration as an integrated system. The IMU was calibrated in situ on the hand to account for its placement and orientation within the glove, while the RFS sensors were simultaneously aligned using the pre-mapped resistance-to-angle correlation from above. This approach ensured that the entire system operated cohesively during motion tracking tests.”
Comment 6: “How do the reported errors (e.g., MAE for the IMU and RFSs) compare to commercially available gloves
Response: Thank you for pointing this out. To address this, we have edited the results section in the manuscript comparing the performance of our glove to similar systems discussed in academic literature. This includes a detailed comparison of the MAE values for the IMU and RFSs with those reported in prior studies, highlighting where our glove performs comparatively well. However, commercially available gloves often do not publish specific error metrics such as MAE. Instead, manufacturers typically provide generalized claims of high accuracy without detailed quantitative data. This lack of transparency makes direct performance comparisons with commercial gloves challenging.
The included content is as follows (Pages 10 and 11):
“For instance, when compared to the glove designed and validated by Connolly et al. their mean error for IMU sensors were 5.95° [38]. Hazman et al. similarly had a IMU glove with a static mean error of 5.41° as did Mohan et al. with an IMU glove with a MAE of 6.3° for wrist flexion and extension [39, 6].”
“The MAE for static finger conditions for the 4 fingers were generally lower than other RFS based gloves, however, the thumb data was consistent with other literature. For instance, in Gentner and Classen’s low-cost sensor glove, the overall error of the RFSs was reported as 4.96° which is higher than the error found in this glove [40]. Other RFS-based gloves have errors of 3.4°.”
Comment 7: “What improvements are planned for glove design? For example, you could include the Iot concept, which will be beneficial for tele-rehabilitation.
Response: Thank you for your comment. There are some improvements planned for the glove in the future. This will be explained in future works but to make that clear in this works there will also be a slight change to the limitation section of the paper. In response to your feedback, we have added a section in the manuscript that addresses this concept and highlights how incorporating wireless connectivity and real-time data transmission could enhance the glove's functionality. This addition outlines the benefits of enabling data sharing with healthcare providers, which could support tele-rehabilitation and reduce the need for in-person consultations. While this section addresses a significant improvement, other advancements, such as additional ergonomic refinements or expanded functionalities, are beyond the scope of this study. These will be explored in future research.
The included content is as follows (Page 14):
“To enhance the functionality of this glove, subsequent research should incorporate wireless connectivity. This will encompass the integration of the Internet of Things paradigm with the transmission of data in real-time. This advancement will enhance the glove's utility, enabling the transmission of data to healthcare providers for tele-rehabilitation, thereby eliminating the necessity for in-person consultations.”
Comment 8 and 9: “Some typos should be corrected for example in the abstract: ''It is be especially...." ?? Some sentences could be more concise. For example, "The glove’s performance was tested with a novel angle testing setup mounted on a collaborative robot" might be shortened”
Response: Thank you for pointing out these areas for improvement. We have carefully reviewed the abstract and corrected the typo, ensuring the text now reads clearly. Additionally, we have revised the suggested sentence to make it more concise. Beyond these specific changes, we conducted a thorough review of the manuscript to identify and correct any remaining typos or lengthy sentences. We believe this has significantly improved the overall clarity and readability of the paper.
Round 2
Reviewer 1 Report
Comments and Suggestions for Authors
The authors have properly answered all my comments.